# An integrated multi-criteria decision-making model for long-term planning of UAVs in disaster management

**Mustafa Erdem Bakir**[1], **Fatih Kasimoglu**[2]*

1 Department of Unmanned and Autonomous System Engineering, Graduate School of Natural and Applied Sciences, University of Turkish Aeronautical Association, Ankara, Turkey, 2 Department of Industrial Engineering, University of Turkish Aeronautical Association, Ankara, Turkey

* ffatihkasimoglu@gmail.com, fkasimoglu@thk.edu.tr

## Abstract

With their superior capabilities, unmanned aerial vehicles (UAVs) play a crucial role in search, rescue, and surveillance operations in disaster management. It is of great importance in the long run to optimally designate the base locations and deployment plans of the UAVs needing a base for their operations. In this study, we develop an integrated multi-criteria decision-making model to select bases and plan missions of UAVs using a combination of multi-attribute and multi-objective optimization techniques, with the decision maker having an interactive role. We formulate a goal programming model in which the number of bases, flight distance, unairworthy days, and cost are jointly minimized. The Analytic Hierarchy Process (AHP) is used to designate the associated goal weights. We develop Algorithm 1 to identify the target level for each goal and Algorithm 2 to refine the model for better solutions. We apply the process in a problem setting where designated disaster activity zones (DAZs) need to be covered by some candidate bases, among which an optimal selection is made. The model's validation and refinement were evaluated across multiple scenarios. The illustrative example yields improvements of 8.57% in cost in the first scenario and 7.54% in distance in the second. The third scenario achieves 7.66% and 6.58% improvements in distance and cost, respectively. A real-world earthquake scenario from Türkiye further demonstrates the model's practical applicability, with 5% improvement in distance and 14.8% in cost. The results of the proposed decision-making process guarantee satisfactory solutions for long-term base and operational planning of UAVs.

## 1. Introduction

Unmanned aerial vehicles (UAVs) stand out as one of the fastest-developing technologies of our time. They are attractive systems because they eliminate the risk of human loss, are low in cost, and provide fast and effective surveillance over large

**Data availability statement:** Data is available from https://github.com/fkasimoglu/Integrated-MCDM-Model-for-UAVs.

**Funding:** The author(s) received no specific funding for this work.

**Competing interests:** The authors have declared that no competing interests exist.

and risky areas. With their superior capabilities, they play a crucial role in search, rescue, and surveillance operations in disaster management. Having a leverage effect in many complicated tasks, they provide a new vision for even disaster management [1] and are successfully used in prediction, assessment, and response processes of disasters such as forest fires [2], earthquakes [3], hurricanes [4], etc. As the missions in which UAVs are involved expand, the processes related to their operations become more complex, making it necessary to develop more sophisticated decision-making processes. One complex and critical decision concerning the operation of UAVs is to optimally designate the base locations of UAVs and make a deployment plan that will accomplish the mission using minimum resources. The UAVs used for long-range distances for search, rescue, and surveillance operations in disaster management essentially need a base to take off and land.

The literature on the selection of the base locations and operation planning of UAVs needing a base is quite limited. In one of the studies, Yakıcı [5] studied the location and routing problem in UAVs, considering the small ships as platforms or bases for UAVs. Chauhan et al. [6] work on choosing possible drone take-off locations with maximum coverage according to the requested location and the number of drones. Akram et al. [7] tried to find the minimum number of drones as flying base stations in a disaster area while maximizing the number of users given service. Liu et al. [8] developed a binary integer programming model for the optimization of base locations and patrol routes in border surveillance. In another paper, Olgac and Toz [9] studied the optimum locations of ground control stations of UAVs operating in a specified search and rescue area. Finally, Park et al. [10] and Claessen et al. [11] studied the location selection and allocation of unmanned aerial vehicles in healthcare applications.

The aforementioned studies have notable contributions to the literature in terms of UAV base locations and their use. However, in none of these studies are the preferences of a decision maker (DM) integrated into the results and the decision-making process. Most of the time, the gap between theory and practice makes it necessary to get DM involved in the process and get the proposed model validated accordingly. In this study, keeping DM in the loop, we develop an original decision-making process with a new approach in which we use a combination of multiattribute and multiobjective techniques to optimally designate the UAV base locations and deployment plan in search, rescue, and surveillance operations in disaster management. Thus, we formulate a goal programming model, in which the number of bases, flight distance, unairworthy days, and cost are designated as the goals for DM. Then, putting DM in the loop, we develop a decision-making model for UAV base selection and deployment plan, and obtain a validated model through the refinement process in line with the interactive feedback received from DM. In this regard, our main contributions can be given as follows.

- For the first time, a decision-making model with DM having an interactive role in the solution process to designate UAV base locations and deployment plan is made available.

- An original integrated multi-criteria decision-making model combining both multiattribute and multiobjective techniques is developed to better represent real-life cases.

- A validated model refined through the interactive feedback from DM is obtained.

- A solution with minimal deviations from the designated goals, ensuring DM's satisfaction, is presented.

The remaining sections of the paper are organized as follows. In Section 2, we mention some basic concepts for the main methods used in the study, namely multi-objective optimization, goal programming, and analytic hierarchy process. We develop our mathematical models and propose a decision-making process in Section 3. The numerical results we get in our application and the discussion are presented in Section 4. Finally, we give our conclusions and propose potential areas for future work in Section 5.

## 2. Materials and methods

### 2.1. Multi-objective optimization

In single-objective model approaches, a single goal is determined, and the optimum value is tried to be reached. However, in real-world problems, decision makers want to optimize many objectives simultaneously. In multi-objective models, sometimes there may be a conflict between defined objectives. In other words, while trying to optimize one goal, there may be a degradation in the desired value of the other goal. If the value of a specific objective function cannot be improved without degradation of one or more objectives, the solution is said to be efficient or Pareto optimal [12].

There are several methods used in applications to solve multi-objective problems, with the most common ones being the weighted sum method [13], compromise programming [14], goal programming [15], prioritized/ lexicographic approaches [16], and epsilon constraint method [17]. In our decision-making model, we use goal programming to obtain efficient solutions to the developed multi-objective model since the designated goals enable the decision maker to have elasticity in his/her preferences and get an interactive role in the process.

### 2.2. Goal programming

Goal programming first appeared in the 1950s [18]. Since then, the topic has been extensively covered with considerable improvements both in theory and practice. Extensive bibliographic studies on goal programming can be found in Romero [19]. Goal programming can be considered as a branch of multi-objective optimization, which itself is a part of multicriteria decision analysis [20]. The aim of goal programming is to minimize deviations from the determined target values. In this way, effective and satisfactory solutions are provided even if not all objectives are optimized for the problem. Rardin [21] presents the formulations of goal programming problems in three categories. The first approach is in weighted goal programming. In this approach, goals are assigned weights based on their relative importance, and then a solution is found that minimizes the weighted sum of deviations from the goals. The second approach is preemptive or lexicographic goal programming. In this method, the goals or objectives are optimized one by one, starting from the most important one. By preserving the optimality of the most important goal, the second most important goal is optimized, and so on. Finally, the third one is preemptive goal programming by weighting the objective, which is a combination of weighted goal programming and preemptive goal programming.

There are many different application areas in which goal programming is effectively used as a multi-objective method. Some notable ones include supplier selection [22,23], portfolio selection [24,25], identifying firefighting strategies [26], resource allocation [27], locating emergency medical service facilities [28], and supply chain management [29–31].

### 2.3. Analytic hierarchy process

When the literature is examined, it is seen that the analytical hierarchy process (AHP) is one of the most used multi-attribute decision-making (MADM) techniques. In this method, first of all, criteria and their sub-criteria are

determined, and a hierarchical structure is created. Pairwise comparisons are made between the identified decision criteria, and the principles of mutuality, homogeneity, dependence, and expectation are used to prioritize each criterion [32]. In creating pairwise comparison matrices, the importance scale between 1–9 suggested by Saaty [33] is used. With the help of the created matrices, the priority value for each criterion is found. The sum of these values equals 1. The criterion with the highest value is the most critical one for the given problem. In this way, the method creates a powerful and easy-to-understand method that allows combining objective and subjective factors in the decision-making process.

The analytic hierarchy process has been used in many different fields that require finding the best alternative considering the multiple criteria influencing the decision process. The examples include software selection [34–36], personnel employment [37–39], supplier evaluation [40–42], safety [43–45], and risk [46,47] assessments, as well as determining treatment alternatives in medication [48].

## 3. A goal programming model

### 3.1. Defining the problem

UAVs used in long ranges with heavy payloads require a base to take off and land for their operations. To efficiently operate UAV deployments, it is important to designate the locations of the bases and plan the missions from the bases to the disaster activity zones (DAZs) in an optimal way. Suppose that there are *I* number of potential base locations and *J* number of disaster activity zones or points requiring UAV deployments/missions in a certain area. As an example, Fig 1 shows how possible service stations (candidate base locations) and demand points (disaster activity zones) are distributed in a certain area of interest with 20 candidate base locations and 7 disaster activity zones.

Each of the existing bases has its own capacity and shortcomings concerning the operational use of UAVs. Some certain bases may have hangars and tents required for the deployment of unmanned aerial vehicles. There is an establishment cost incurred to use a base for UAVs since they are normally designed for the usual civil and/or security use. The safe zone covers a large area, which is why meteorological conditions differ in the base locations, with an expected number of unairworthy days in a year. The aim is to optimally determine the base locations of operative-level unmanned aerial vehicles for flight missions from the safe zone to risky zones, considering specific mission requirements and costs as well. The following key concepts are essentially used in the development of our mathematical models for the problem.

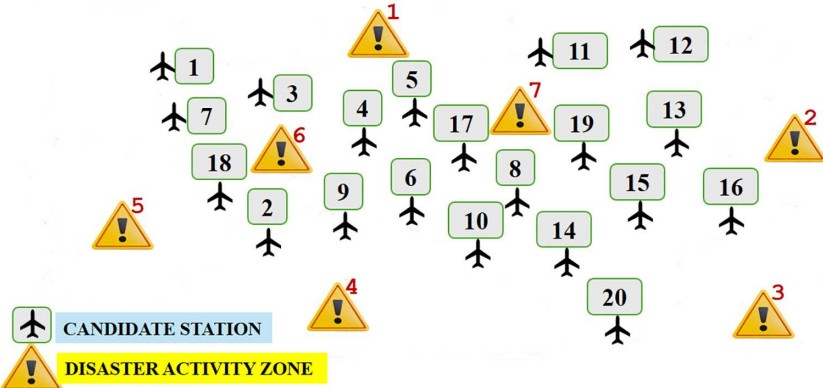

**Fig 1. Area of interest, candidate base stations, and disaster activity zones.**

There is a setup cost incurred to use a base for UAVs. Choosing a candidate base for unmanned aerial vehicles for the first time involves some new adjustments in the base before it is ready for an operation. These amendments in the base include the construction of hangars, tents, living spaces, the supply of fuel tankers and other logistics materials, requiring some fixed costs to make it eligible for UAV usage. Fuel consumption cost, on the other hand, constitutes a variable cost depending on the flight distances from the selected bases to the DAZs.

The area of interest covers a huge area, in which meteorological conditions differ in the base locations, with an expected number of unairworthy days in a year. The total weight of unmanned aerial vehicles is relatively less than other aerial vehicles. Therefore, UAVs are more affected by meteorological conditions during their operations, especially during take-offs and landings. Particularly, thunderstorms, strong wind conditions, and heavy cloudiness are unfavorable conditions for the UAVs' flight. The number of days with adverse weather conditions for candidate bases is obtained from the meteorological data in the base location.

The expected number of needed operational missions naturally differs in a designated DAZ based on the experiences regarding the intensity of the incidents in the past, the density of the population and forests, the intensity of flood-risky watercourses, and seismic activities in the region. We assume that the number of flight missions needed from a base to a DAZ is available.

In this regard, a decision maker is to designate the bases from which the UAVs will be deployed to the DAZs. The problem can be considered as a facility location problem. Typically, in a facility location problem, the DM selects a subset of facilities to provide service to all service or demand points with an objective to minimize cost [49]. When the budget is not a concern at all, and the demands must be completely met, as in the set covering problems, the objective is to minimize the number of facilities fully fulfilling the demand requirements [50]. The existence of multiple conflicting objectives in a facility location problem and the search to better utilize the resources make it necessary to apply a multiobjective approach for the solution [51]. In our study, in addition to minimizing the cost and the number of bases (facilities), we designate two more objectives, the number of unairworthy days in an air base and total flight distance, which are two critical elements to be minimized in UAV operations. The weather conditions must be convenient for landing and take-off for UAVs needing an airbase, and in a disaster case, the flight distance, which determines the response time, must be kept to a minimum.

Focusing on the long-term planning of UAV operations, with particular emphasis on the selection and designation of appropriate bases, the main assumptions from an operational perspective are outlined as follows.

- There is a sufficient number of well-established bases within the area of interest from which an appropriate subset can be selected for the UAVs under consideration, consistent with typical facility location problems.

- The UAVs operate at high altitudes, use mainly satellite communication, have substantial observational capacity with high-tech imaging systems, and are capable of extended flight durations.

- The weather data specific to the region where an existing air base is located is available and indicates the expected number of unairworthy days.

### 3.2. Goal programming model

*Notation*

Indices:

 $k$: goals to achieve *(k=1,…,K)*

 $i$: candidate UAV bases *(i=1,…,I)*

 $j$: disaster activity zones (DAZs) *(j=1,…,J)*

Parameters:

$G_k$: target achievement level of goal $k$

$w_k$: weight for the deviation in the targeted level of goal $k$

M: an arbitrarily big positive number

$FD_j$: number of flight demand in DAZ $j$

$R_{ij}$: distance from base $i$ to DAZ $j$ (radius in km)

$RA_{ij}$: altered distance from base $i$ to DAZ $j$ for binary calculations

$$\begin{cases} 0, & \text{if the distance from base } i \text{ to DAZ } j \text{ surpasses the maximum allowable distance} \\ MR_{ij}, & \text{otherwise} \end{cases}$$

$F_i$: fixed cost of using base $i$ for UAVs (in dollars)

C: cost of flying an unmanned air vehicle per km (in dollars)

$UAD_i$: expected number of unairworthy days at base $i$

$Cap_i$: flight capacity of base $i$ regarding the number of missions

$T_k$: proportional threshold deviation for goal $k$

Decision Variables:

$y_i$ : $\begin{cases} 1, & \text{if base } i \text{ is selected} \\ 0, & \text{otherwise} \end{cases}$

$x_{ij}$: annual number of UAV deployments/missions planned from base $i$ to DAZ $j$

$d_k$: deviation in the targeted achievement level of goal $k$

Z: weighted total proportional deviation from the targeted achievements (obj. function value)

We develop the following goal programming model, Model 1, to select and figure out the annual number of UAV deployments from each base $i$ to each DAZ $j$.

Model 1:

$$\min Z = \sum_{k=1}^{K} = w_k \left( \frac{d_k}{G_k} \right) \tag{1}$$

Subject to:

$$\sum_{i=1}^{I} y_i \leq G_1 + d_1 \tag{2}$$

$$\sum_{i=1}^{I} \sum_{j=1}^{J} R_{ij} x_{ij} \leq G_2 + d_2 \tag{3}$$

$$\sum_{i=1}^{I} UAD_i y_i \leq G_3 + d_3 \tag{4}$$

$$\sum_{i=1}^{I} \sum_{j=1}^{J} CR_{ij} x_{ij} + \sum_{i=1}^{I} F_i y_i \leq G_4 + d_4 \tag{5}$$

$$\sum_{i=1}^{I} x_{ij} \geq FD_j \qquad (j = 1, \ldots, J) \tag{6}$$

$$\sum_{j=1}^{J} x_{ij} \leq Cap_i \, y_i \qquad (i = 1, \ldots, I) \tag{7}$$

$$x_{ij} \leq RA_{ij} \qquad (i = 1, \ldots, I), \ (j = 1, \ldots, J) \tag{8}$$

$$y_i = 0 \ or \ 1 \qquad (i = 1, \ldots, I) \tag{9}$$

$$x_{i,j} \geq 0 \qquad (i = 1, \ldots, I), \ (j = 1, \ldots, J) \tag{10}$$

$$d_k \geq 0 \qquad (k = 1, \ldots, K) \tag{11}$$

In Model 1, the objective function (1) minimizes the total proportional deviations ($\frac{d_k}{G_k}$) from the targeted goal levels in line with the identified weights obtained from a decision maker via AHP. Constraints (2), (3), (4), and (5) represent the four designated goals relating to the minimum number of bases, total flight distance, total annual unairworthy days, and total cost, respectively. Note that each goal is given with some deviation, $d_k$. Constraint (6) is used to meet the flight mission demands at each DAZ $j$. Constraint (7) ensures that the annual flight capacity is not exceeded at each base. Constraint (8) allows missions only to distances less than the designated maximum value. Constraints (9) and (10) are used to define the variables.

### 3.3. Algorithm to find target achievement levels ($G_k$'s)

Defining a realistic and achievable target value for a goal can be complicated due to the complexity of the problem and the existence of multiple conflicting goals. We provide Algorithm 1 to find the best value of each goal if only a single goal were to be observed under the given constraints of the problem. The found values are used as a target value, $G_k$, for each of the goals.

**Algorithm 1:**

```
Read Parameters
Assign a value of 0 to all w_k parameters
{for k=1 to K do
w_k = 1
Solve Model 1
G_k = optimal Z value of Model 1
w_k = 0}
```

Note that in Algorithm 1, each time Model 1 is solved by setting $w_k = 1$ for the goal $k$ in concern and $w_k = 0$ for the other goals to ensure that Model 1 finds the best achievable value for each of the defined goals.

### 3.4. Algorithm to validate and refine the model

Even though Model 1 finds an optimal solution, in real life, the results might not satisfy DMs due to the gap between theory and practice and the inherent subjectivity of the evaluation of the goal weights by DM. Integrating DM's feedback through the process in an interactive way, we propose Algorithm 2 to get a refined and validated model before its implementation.

**Algorithm 2:**
```
Get threshold goal deviation values, T_k's, from DM
Read Parameters
{for k=1 to K do
```
*Add the new constraint*, $\frac{d_k}{G_k} \leq T_k$, *to Model* 1 }
```
Solve updated Model 1
```

Algorithm 2 enables DM with a solution better reflecting the DM's real preferences through integrating the threshold value ($T_k$) obtained from DM for the goal achievement levels that do not fully satisfy DM.

### 3.5. Integrated MCDM process

The steps of the multi-criteria decision-making model are shown in Fig 2, with DM having an interactive role both in multi-attribute and multi-objective decision-making processes.

Steps 1, 2, and 3 constitute the multi-attribute decision-making phase, whereas Steps 4,5,6,7, and 8 relate to the interactive multi-objective decision-making procedure of the integrated decision-making process proposed in Fig 2. In this respect, at Step 1, DM is asked to make a pairwise comparison between the identified goals in accordance with the AHP process; at Step 2, goal deviation weights, $w_k$ values, are calculated; at Step 3, the consistency of DM is checked in an iterative manner to ensure that the calculated $w_k$ values are reliable.

Once $w_k$ values are decided, the multi-objective decision-making process starts employing basically goal programming in an interactive way with DM. At Step 4, Algorithm 1 is applied to calculate the target achievement level, $G_k$, for each goal. At Steps 5 and 6, Model 1 is solved, and the solutions are presented to DM, respectively. Depending on how DM is satisfied with the results, the process moves either to Step 7, in which the decision is made, and the implementation is on, or to Step 8, in which the model is refined through Algorithm 2 with the feedback from DM for those achievement levels DM is not sufficiently satisfied with. At this step, the model is modified with the new input from DM, and the solution is once again offered to DM at Step 6.

As a result, the model is refined through repetitive feedback from DM, and a more validated version of the model is attained. Since the process outlined in Fig 2 continues until the DM stops giving feedback for the solution outcomes, a satisfactory solution is guaranteed.

## 4. Application results and discussion

In this part of the study, we first apply the developed integrated process in Fig 2 step by step and present the resulting outcomes in an illustrative example. We then present a real-world scenario regarding the case of an earthquake in Türkiye.

### 4.1. An illustrative example

**4.1.1. Determining goal deviation weights using AHP (Steps 1, 2, and 3).** We apply AHP to determine the weight for each goal, associated $w_k$ value. Thus, at Step 1 of the integrated model, we construct our pairwise comparison matrix

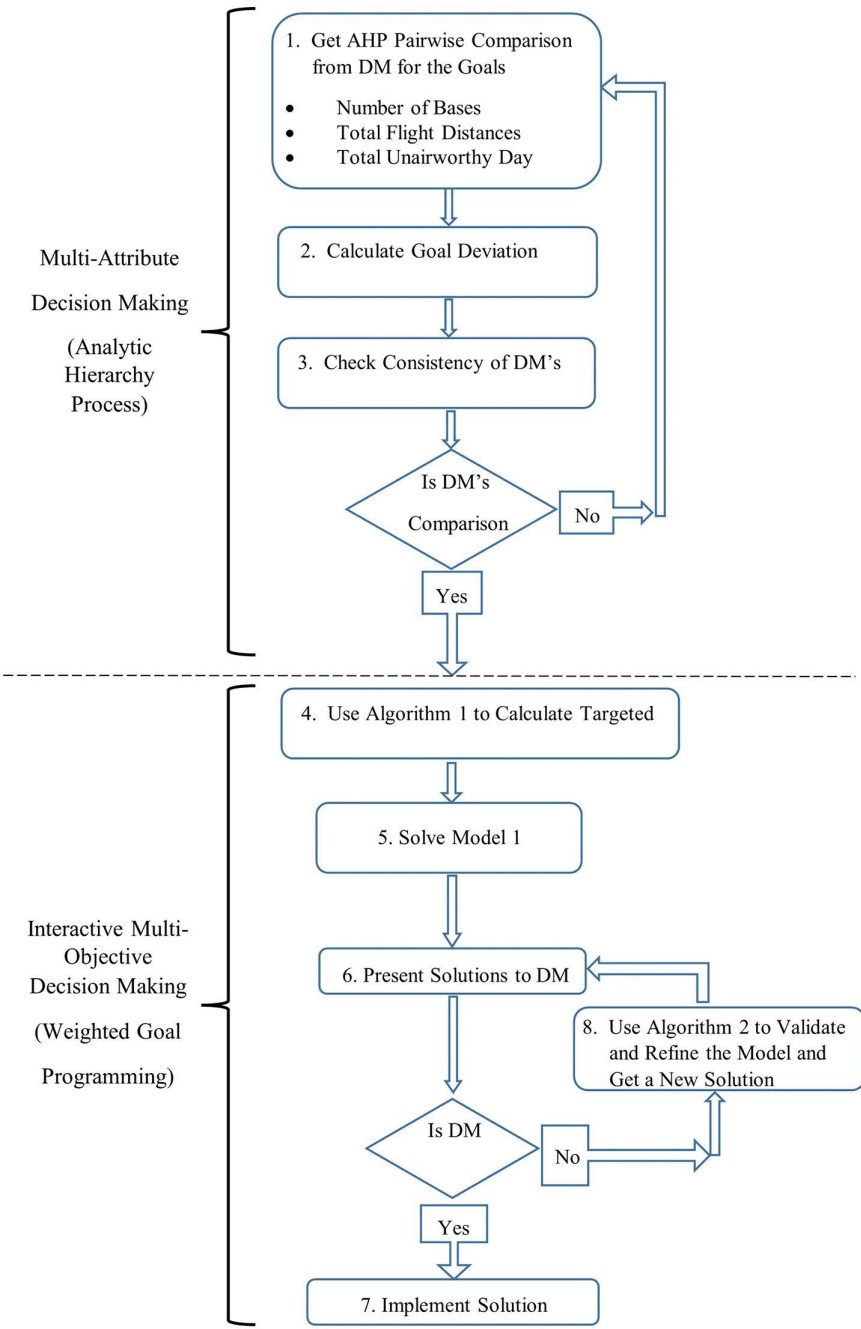

**Fig 2. Integrated multi-criteria decision-making model for planning of UAVs.**

reflecting the assessment of a decision maker regarding how important the goals are when compared to each other, as given in Table 1.

Our normalized comparison matrix, together with the calculated goal weights and consistency measures, is given in Table 2 as an outcome of Step 2 and Step 3 of the model. As a result, we use $w_k$ values, 0.1017, 0.2776, 0.5696, and

**Table 1. Pairwise comparison matrix.**

|  | Number of Bases | Flight Distance | Unairworthy Days | Cost |
|---|---|---|---|---|
| Number of Bases | 1 | 0.3333 | 0.2000 | 2 |
| Flight Distance | 3 | 1 | 0.3333 | 7 |
| Unairworthy Days | 5 | 3 | 1 | 9 |
| Cost | 0.5000 | 0.1429 | 0.1111 | 1 |

**Table 2. Normalized comparison matrix, goal weights, and consistency measures.**

|  | Number of Bases | Flight Distance | Unairworthy Days | Cost | Goal Weight ($w_k$) | Consistency Measure |
|---|---|---|---|---|---|---|
| Number of Bases | 0.1053 | 0.0745 | 0.1216 | 0.1053 | 0.1017 | 4.0379 |
| Flight Distance | 0.3158 | 0.2234 | 0.2027 | 0.3684 | 0.2776 | 4.0735 |
| Unairworthy Days | 0.5263 | 0.6702 | 0.6081 | 0.4737 | 0.5696 | 4.1632 |
| Cost | 0.0526 | 0.0319 | 0.0676 | 0.0526 | 0.0512 | 4.0041 |
|  |  |  |  | Consistency Index (CI): | 0.0232 | |
|  |  |  |  | Consistency Ratio (CR): | 0.0258 | |

0.0512 for our four goals, number of bases, flight distance, unairworthy days, and cost, respectively. The $w_k$ values indicate that the goal associated with unairworthy days has the largest weight, 0.5696, while the cost has the least weight value, 0.0512. Flight distance has the second largest weight with a value of 0.2776, and next comes the number of bases with a weight value of 0.1017, as seen in Table 2. Note that the consistency of DM in his/her comparison is verified with a consistency ratio of 0.0258 (≤ 0.1) at Step 3.

**4.1.2. Obtaining initial solution (Steps 4 and 5).** In this part of the study, we solve our goal programming model, Model 1, in its original form with generic data using GAMS [52] release 24.4.1 and CPLEX 12.6.1.0. As described in Fig 2, we apply Algorithm 1 at Step 4 to get the associated target achievement value, $G_k$, for each goal. Then, with the found $G_k$ values we solve Model 1 at Step 5 using the $w_k$ values calculated in Section 4.1. Table 3 summarizes the obtained results, and Fig 3 shows the optimal base locations as well as the allocated DAZs receiving service from these base locations.

As seen in Table 3, 8 bases are selected among 20 candidate bases to carry out the flight missions requested by 7 DAZs. DAZs where UAVs are deployed to, and the number of flight missions from the opened bases to these zones, are also given in Table 3. It can be seen from the table that the mission requirement of DAZ 1 is met from Base 3 and 5. Likewise, the requests of DAZ 2 are met from Base 16 and 20, the requests of DAZ 3 are met from Base 10 and 20, the requests of DAZ 4 are met from Base 9 and 10, the requests of DAZ 5 are met from Base 2 and 18. The annual number of flight missions planned from each selected base to each DAZ is also given in Table 3. As an example, Zone 1 gets 769 flight missions from Base 3 and 1056 flight missions from Base 5.

Model 1 results regarding the achieved levels of goals are presented in Table 4. When Table 4 is examined, we see that achieved goal levels for the number of bases, total flight distance, total unairworthy days, and total cost turn out to be 8, 4792590, 833, and 12580890, respectively. The associated percentage deviations are observed as 0, 18.4, 2, and 18.9.

The deviation in the total cost is noticeably high with a value of 18.9 percent. The reason for this is that DM attaches the lowest importance to the cost criterion in the pairwise comparison process discussed in section 4.1.

**4.1.3. Validation and refinement of the model (Steps 6, 7, 8).** In accordance with the proposed decision-making process in Fig 2, Step 6, we try to validate our model by presenting the results to DM and integrating his/her feedback and satisfaction level into the process. In this respect, we consider three different cases described as follows.

**Table 3. Model 1 solution results.**

**Optimal Objective Function Value (Minimum Total Weighted Goal Deviation) = 7.2%**

| Number of Selected Bases | Selected Bases | Allocated Disaster Activity Zones | AnnualFlight Missions |
|---|---|---|---|
| 8 | Base 2 | Zone 5 | 729 |
| | | Zone 6 | 303 |
| | Base 3 | Zone 1 | 769 |
| | | Zone 6 | 311 |
| | Base 5 | Zone 1 | 1056 |
| | Base 9 | Zone 4 | 958 |
| | | Zone 6 | 106 |
| | Base 10 | Zone 3 | 144 |
| | | Zone 4 | 122 |
| | | Zone 7 | 720 |
| | Base 16 | Zone 2 | 924 |
| | Base 18 | Zone 5 | 1096 |
| | Base 20 | Zone 2 | 516 |
| | | Zone 3 | 576 |

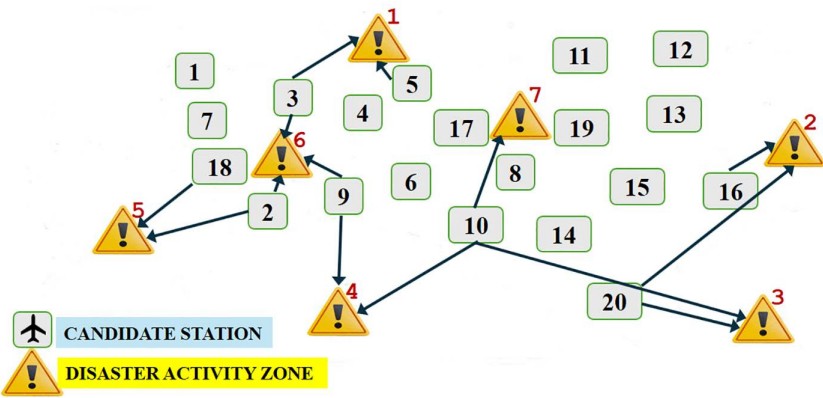

**Fig 3. Optimal base locations and allocated DAZs according to Model 1.**

**Table 4. Targeted and achieved levels of goals obtained from Model 1.**

| Objectives/ Goals | Targeted Level | Achieved Level | Deviation in Targeted Level | Percentage Deviation in Targeted Level |
|---|---|---|---|---|
| **Number of Bases Opened** | 8 | 8 | 0 | 0% |
| **Total Flight Distance (km)** | 4046475 | 4792590 | 746115 | 18.4% |
| **Total Unairworthy Days** | 817 | 833 | 16 | 2% |
| **Total Cost (dollars)** | 10582690 | 12580890 | 1998197 | 18.9% |

*Case 1: DM is not satisfied with the resulting cost value and sets a threshold value for the deviation from the targeted cost value*

In our first scenario, DM considers the deviation value of 18.9% in Goal 4, total cost, as being too much and not satisfactory. Thus, DM sets a threshold value of 15% for the deviation from Goal 4. To integrate the feedback from DM into the model and get a more validated and refined one, we employ Algorithm 2 at Step 8 with the following new constraint (12) added to Model 1.

$$\frac{d_4}{G_4} \leq 0.15$$

(12)

The solution obtained through Algorithm 2 is summarized in Table 5 with the optimal base locations depicted in Fig 4.

**Table 5. The refined model Case 1 solution results.**

| Optimal Objective Function Value (Minimum Total Weighted Goal Deviation) = 7.8% | | | |
|---|---|---|---|
| Number of Selected Bases | Selected Bases | Allocated Disaster Activity Zones | AnnualFlight Missions |
| 8 | 2 | Zone 5 | 729 |
| | | Zone 6 | 303 |
| | 3 | Zone 1 | 777 |
| | | Zone 6 | 303 |
| | 4 | Zone 1 | 1048 |
| | 9 | Zone 4 | 950 |
| | | Zone 6 | 114 |
| | 10 | Zone 3 | 144 |
| | | Zone 4 | 130 |
| | | Zone 7 | 720 |
| | 16 | Zone 2 | 924 |
| | 18 | Zone 5 | 1096 |
| | 20 | Zone 2 | 516 |
| | | Zone 3 | 576 |

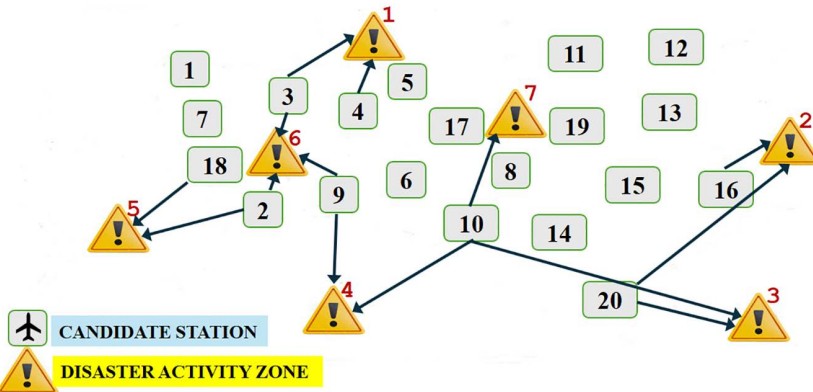

**Fig 4. Optimal base locations and allocated DAZs according to the refined model: Case 1.**

When we examine Table 5 and Fig 4, we can observe the changes in the selected bases and number of allocated missions to the DAZs in the refined model when compared to the original Model 1 results in Table 3, even though the number of selected bases remains the same as before 8. Considering the selected bases in Table 5, we see that unlike Model 1 results in Table 3, Base 5 does not exist among the selected bases. Base 4, on the other hand, is seen in the solution as one of the selected bases in Table 5. In addition, we observe that the flight missions to be carried out from the selected bases to the DAZs have also changed in the refined model. For example, we observe that Base 3 is a selected base both in Tables 3 and 5 with the same allocated DAZs, Zone 1 and Zone 6. However, the numbers of annual flight missions from this base to Zone 1 and Zone 6 are 769 and 311, respectively, in Table 3, whereas the same numbers turn out to be 777 and 303 in Table 5.

Results of the refined model relating to the targeted and achieved levels of goals are given in Table 6. The achieved goal levels for the number of bases, total flight distance, total unairworthy days, and total cost turn into 8, 4937910, 835, and 11501920, respectively. The associated percentage deviations from targeted levels are found to be 0, 22, 2.2, and 8.7. We observe a noticeable improvement of 8.57% in the achieved cost value when compared to that of the original Model 1 results in Table 4, a decrease of $1078970 from $12580890 to $11501920, even though some increases in other deviational values are observed. The deviation from the targeted level of cost drops from 18.9% to 8.7%. The model offers an option that better reflects DM's aspirations with the threshold value received from DM. Thus, the refined model with DM in the loop gives more validated and satisfactory results with considerable improvement in the achieved goal level for which DM gives his/her feedback.

*Case 2: DM is not satisfied with the resulting flight distance value and sets a threshold value for the deviation from the targeted flight distance value*

In this scenario, DM considers that the deviation value of 18.4% in Goal 2, total flight distance, is too high and not satisfactory. Thus, DM establishes a threshold value of 15% for the deviation in Goal 2. To integrate the feedback, Algorithm 2 is applied with the following constraint (13) added to Model 1.

$$\frac{d_2}{G_2} \leq 0.15$$

(13)

Table 7 summarizes the solution of Algorithm 2, and Fig 5 shows the associated optimal base locations. When we look at Table 7 and Fig 5, we can see the differences in the results when compared to those obtained through the original Model 1 in Table 3. First, it is seen that the number of selected bases in this case increases from 8 to 9. We also notice that Base 10 existing in Table 3 Model 1 solution, does not exist among the selected bases in Table 7 refined model solution. Instead, in Table 7, we see that Base 4 and Base 15 are in the solution, which means that these bases improve the total flight distance value in line with the feedback received from DM. Finally, one should note that there are also differences in the annual flight mission numbers from selected bases to the DAZs when compared to those values in Table 3.

Results of the refined model relating to the targeted and achieved levels of goals are given in Table 8. The achieved goal levels for the number of bases, total flight distance, total unairworthy days, and total cost become 9, 4937910, 835,

**Table 6. Targeted and achieved levels of goals obtained from the refined model: Case 1.**

| Objectives/ Goals | Targeted Level | Achieved Level | Deviation in Targeted Level | Percentage Deviation from Targeted Level |
|---|---|---|---|---|
| **Number of Bases Opened** | 8 | 8 | 0 | 0% |
| **Total Flight Distance (km)** | 4046475 | 4937910 | 891435 | 22% |
| **Total Unairworthy Days** | 817 | 835 | 18 | 2.2% |
| **Total Cost (dollars)** | 10582690 | 11501920 | 919232 | 8.7% |

**Table 7. The refined model Case 2 solution results.**

| Optimal Objective Function Value (Minimum Total Weighted Goal Deviation) =14.4% | | | |
|---|---|---|---|
| **Number of Selected Bases** | **Selected Bases** | **Allocated Disaster Activity Zones** | **AnnualFlight Missions** |
| 9 | 2 | Zone 4 | 16 |
| | | Zone 5 | 729 |
| | | Zone 6 | 287 |
| | 3 | Zone 1 | 13 |
| | | Zone 6 | 433 |
| | 4 | Zone 1 | 1048 |
| | 5 | Zone 1 | 764 |
| | | Zone 7 | 292 |
| | 9 | Zone 4 | 1064 |
| | 15 | Zone 2 | 516 |
| | | Zone 7 | 428 |
| | 16 | Zone 2 | 924 |
| | 18 | Zone 5 | 1096 |
| | 20 | Zone 3 | 720 |

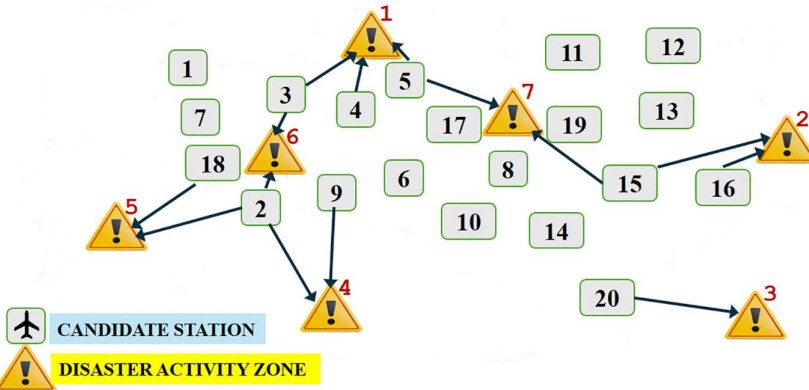

**Fig 5. Optimal base locations and allocated DAZs according to the refined model: Case 2.**

**Table 8. Targeted and achieved levels of goals obtained from the refined model: Case 2.**

| Objectives/ Goals | Targeted Level | Achieved Level | Deviation in Targeted Level | Percentage Deviation from Targeted Level |
|---|---|---|---|---|
| **Number of Bases Opened** | 8 | 9 | 1 | 12.5% |
| **Total Flight Distance (km)** | 4046475 | 4431020 | 384545 | 9.5% |
| **Total Unairworthy Days** | 817 | 951 | 134 | 16.4% |
| **Total Cost (dollars)** | 10582690 | 12895030 | 2312338 | 21.9% |

and 11501920, respectively, for this case. The associated percentage deviations from targeted levels are found to be 12.5, 9.5, 16.4, and 21.9. Note that due to the feedback from DM, there is a remarkable improvement of 7.54% in the achieved flight distance value when compared to that of the original Model 1 solution, a decrease of 361570 km from 4792590 to 4431020, even though some increases in other deviational values are observed. The deviation from the targeted level of flight distance decreases from 18.4% to 9.5%.

*Case 3: DM is not satisfied with the resulting flight distance and cost values and sets a threshold value for both the deviation from the targeted flight distance and the deviation from the targeted cost value*

In this scenario, DM judges that the deviation value of 18.4% in Goal 2, total flight distance, and the deviation value of 18.9% in Goal 4, total cost, are both too much and not satisfactory. Thus, DM sets a threshold value of 15% for the deviation in Goal 2 and for the deviation in Goal 4 as well. To integrate the mentioned feedback from DM, we apply Algorithm 2 with the following constraints (14,15) added to Model 1.

$$\frac{d_2}{G_2} \leq 0.15$$

(14)

$$\frac{d_4}{G_4} \leq 0.15$$

(15)

Applying Algorithm 2, we get the following results given in Table 9, with the optimal base locations shown in Fig 6.

Results of the refined model relating to the targeted and achieved levels of goals in Case 3 are given in Table 10. The achieved goal levels for the number of bases, total flight distance, total unairworthy days, and total cost turn into 9, 4425060, 977, and 11752530, respectively. The associated percentage deviations from targeted levels become 12.5, 9.4, 19.6, and 11.1. The improvements in the distance and cost values are 7.66% and 6.58%, respectively, when compared to those of the original Model 1 results. We observe a decrease with a value of 367530 km from 4792590 to 4425060 in the flight distance and a decrease with a value of 828360 dollars from 12580890 to 11752530 in the cost value.

**Table 9. The refined model Case 3 solution results.**

| Optimal Objective Function Value (Minimum Total Weighted Goal Deviation) = 15.6% | | | |
|---|---|---|---|
| Number of Selected Bases | Selected Bases | Allocated Disaster Activity Zones | AnnualFlight Missions |
| 9 | 2 | Zone 4 | 16 |
| | | Zone 5 | 729 |
| | | Zone 6 | 287 |
| | 3 | Zone 1 | 441 |
| | | Zone 6 | 433 |
| | 4 | Zone 1 | 1048 |
| | 5 | Zone 1 | 336 |
| | | Zone 7 | 720 |
| | 9 | Zone 4 | 1064 |
| | 12 | Zone 2 | 516 |
| | 16 | Zone 2 | 924 |
| | 18 | Zone 5 | 1096 |
| | 20 | Zone 3 | 720 |

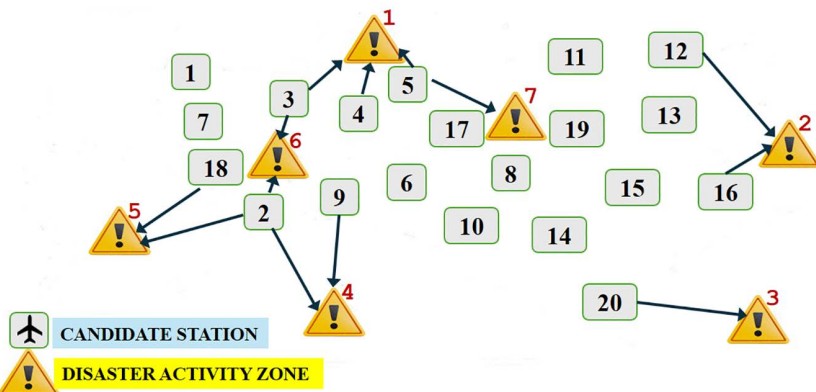

**Fig 6. Optimal base locations and allocated DAZs according to the refined model Case 3.**

**Table 10. Targeted and achieved levels of goals obtained from the refined model Case 3.**

| Objectives/Goals | Targeted Level | Achieved Level | Deviation in Targeted Level | Percentage Deviation from Targeted Level |
|---|---|---|---|---|
| **Number of Bases Opened** | 8 | 9 | 1 | 12.5% |
| **Total Flight Distance (km)** | 4046475 | 4425060 | 378585 | 9.4% |
| **Total Unairworthy Days** | 817 | 977 | 160 | 19.6% |
| **Total Cost (dollars)** | 10582690 | 11752530 | 1169835 | 11.1% |

For comparison, Fig 7 summarizes the percentage deviations obtained in Model 1 and in all three cases. It is noteworthy that, depending on the feedback from DM, relatively low cost values are observed in Cases 1 and 3, and low flight distance values are obtained in Cases 2 and 3. The higher deviation levels in total unairworthy days should also be noted for Cases 2 and 3.

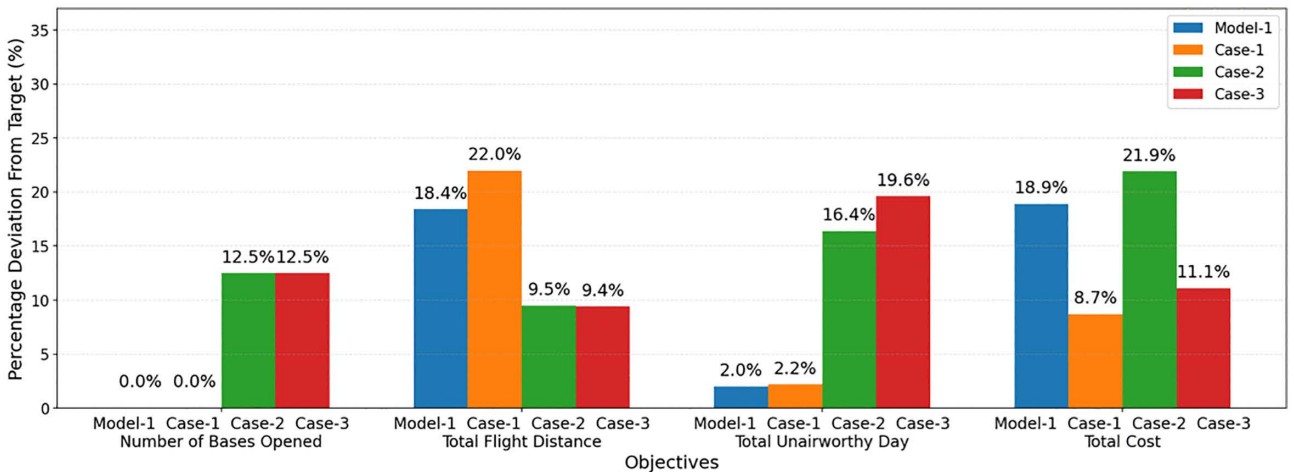

**Fig 7. Percentage deviations in Model 1 and all three cases.**

## 4.2. A real-world scenario: The case of an earthquake in Türkiye

Türkiye is located in a region where the intensity of seismic activity is very high, with several fault lines passing through the country, as can be seen in Fig 8 [53]. According to the United States Geological Survey (USGS) earthquake catalogue [54], 15 devastating earthquakes having a magnitude greater than 6.5 occurred in the last 50 years in the country.

Designating 15 potential disaster activity zones, where there is a high risk of a devastating earthquake, and 29 candidate air bases actively used in the country, as depicted in Fig 9, we apply the solution procedure in a similar way as discussed in Section 4.1. Table 11 summarizes the results derived from Model 1. Fig 10 depicts the eight selected bases across the country from the 29 potential sites. Table 12 presents the corresponding targeted and achieved levels of goals obtained through the solution.

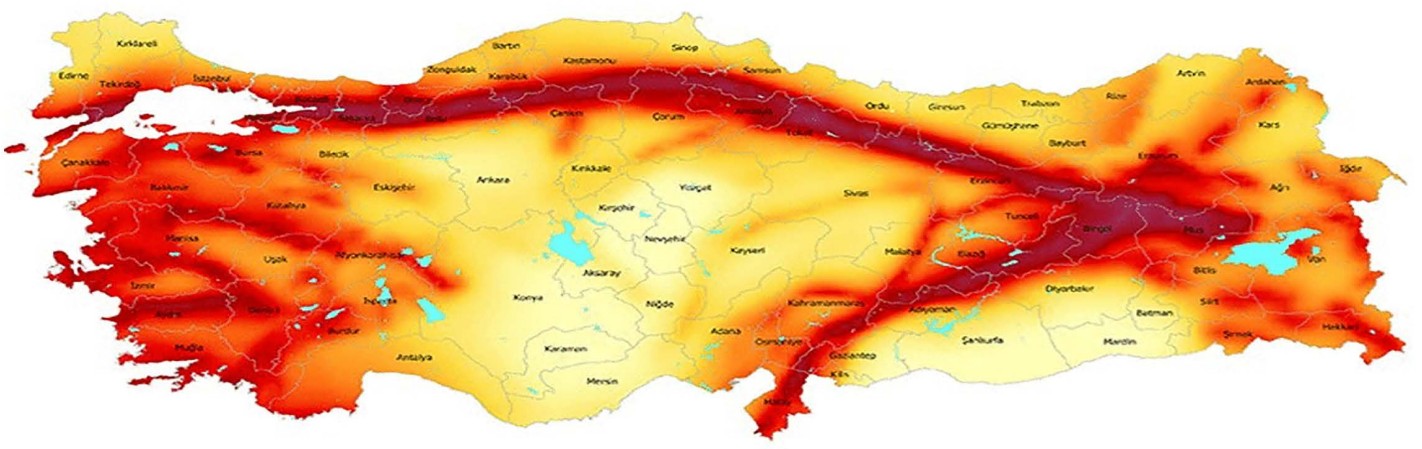

**Fig 8. Fault lines in Türkiye.**

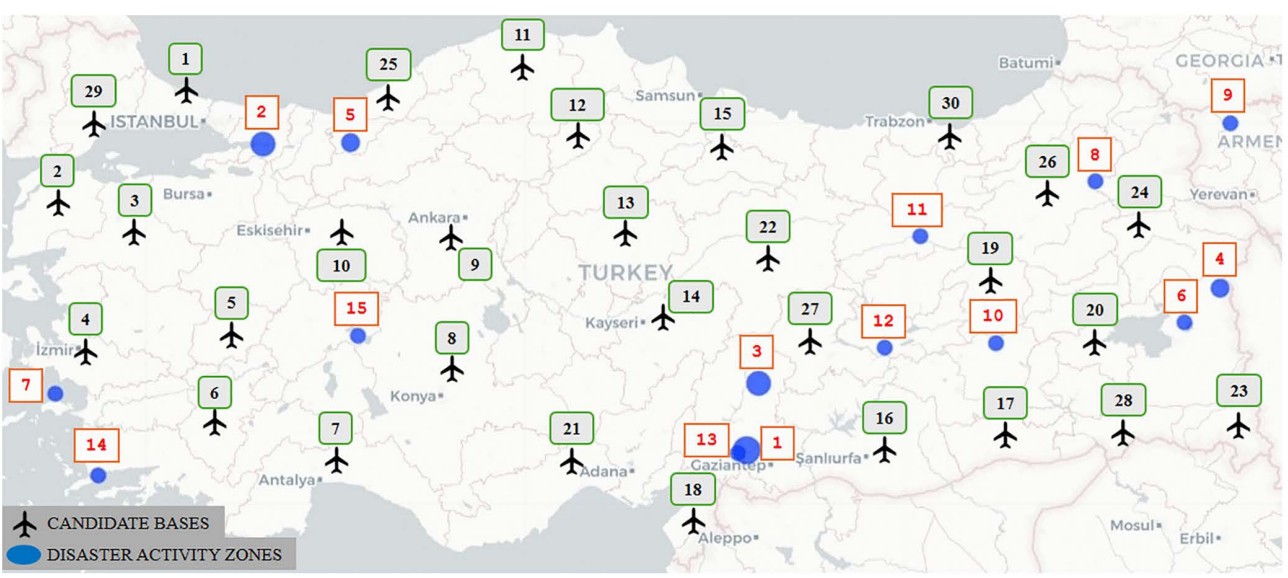

**Fig 9. Candidate bases and disaster activity zones in Türkiye.**

**Table 11. Model 1 solution results in the case of the earthquake scenario.**

Minimum Total Weighted Goal Deviation = 24.7%

| Number of Selected Bases | Selected Bases | Allocated Disaster Activity Zones | Flight Missions Planned |
|---|---|---|---|
| 8 | Base 4 | Zone 7 | 360 |
| | | Zone 14 | 360 |
| | Base 9 | Zone 5 | 720 |
| | | Zone 15 | 76 |
| | Base 10 | Zone 2 | 720 |
| | | Zone 15 | 284 |
| | Base 18 | Zone 1 | 720 |
| | | Zone 3 | 16 |
| | | Zone 13 | 360 |
| | Base 21 | Zone 3 | 704 |
| | | Zone 12 | 336 |
| | Base 23 | Zone 4 | 720 |
| | | Zone 6 | 4 |
| | | Zone 9 | 360 |
| | Base 25 | Zone 8 | 360 |
| | | Zone 11 | 360 |
| | | Zone 12 | 24 |
| | Base 28 | Zone 6 | 716 |
| | | Zone 10 | 360 |

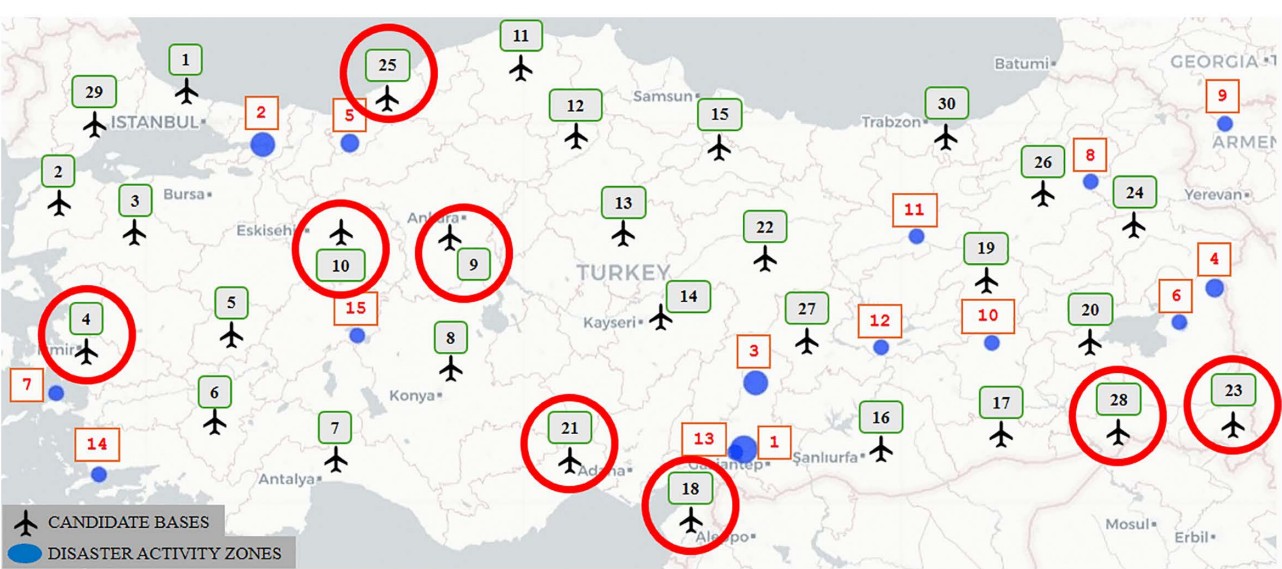

**Fig 10. Optimal base locations for the earthquake scenario as determined by Model 1.**

**Table 12. Model 1 targeted and achieved levels of goals in the case of the earthquake scenario.**

| Objectives/ Goals | Targeted Level | Achieved Level | Deviation in Targeted Level | Percentage Deviation in Targeted Level |
|---|---|---|---|---|
| Number of Bases Opened | 7 | 8 | 1 | 14.3% |
| Total Flight Distance (km) | 717020 | 951900 | 234880 | 32.8% |
| Total Unairworthy Days | 652 | 790 | 138 | 21.2% |
| Total Cost (dollars) | 7777181 | 10967800 | 3190617 | 41% |

Considering that the goal deviations in total flight distance and cost are relatively high, as seen in Table 12, we set a threshold value of 0.30 for both goals to integrate the feedback of a DM and get more refined solutions. The associated refined summary results of the UAV deployment plan, the depiction of the selected bases, and the achieved levels of the goals are presented through Table 13, Fig 11, and Table 14, respectively.

With a threshold value of 0.30% for the deviations in flight distance and cost, we observe the percentage deviation in the targeted flight distance in Table 14 as 27.5% with a decrease of 5% from 32.8% and the deviation in the cost as 26.2% with a decrease of 14.8% from 41%.

## 5. Conclusion and future work

In this study, we develop a multi-criteria decision-making model using a combination of multiattribute and multiobjective techniques to optimally figure out the long-term UAV base location and deployment planning with DM in the loop in an interactive way. We first formulate a goal programming model, Model 1, in which the number of bases, flight distance, unairworthy days, and cost are designated as the goals for a decision maker in the problem. Then, putting DM in the loop, we develop a decision-making model for UAV base selection and deployment plan, and obtain a validated model through the refinement process in line with the interactive feedback received from DM.

In our proposed model, first, DM provides a pairwise comparison between the identified goals to calculate the goal deviation weights, $w_k$ values, and then the consistency of DM is verified in accordance with the AHP process. We develop Algorithm 1 to find the targeted goal level, $G_k$, for each of the goals, which is the best achievable value if only the interested goal is observed. Next, Model 1 is solved, and the solutions are presented to DM. Depending on how satisfactory the results are for DM, the process moves either to the implementation or to the refinement stage that uses Algorithm 2 developed to align the model with the real aspirations of DM through his/her interactive feedback. The new solution is then presented to DM, and another feedback is asked from DM to ensure his/her satisfaction. The repetitive feedback continues until the results are satisfactory enough for DM.

In our application, we apply the proposed model step by step in a problem setting designed with generic data. Obtaining the pairwise comparison matrix from a decision maker, we calculate the goal deviation weights, $w_k$'s, confirm the consistency of DM and determine the targeted goal levels, $G_k$'s. At this stage, we solve the proposed Model 1 and get the achieved levels for our designated goals as follows: 8 for the total number of bases, 4792590 (km) for total flight distance; 833 for total unairworthy days, and $12580890 for the total cost. The resulting percentage deviations are 0, 18.4, 2, and 18.9, respectively, with a total weighted deviation of 7.2 percent. The results also reveal optimal $x_{ij}$ values, the annual number of UAV missions planned from a base $i$ to a DAZ $j$.

To show how the validation and refinement process of our model works, we try three different scenarios in our illustrative example. In our first scenario (Case 1), DM considers the goal deviation in the cost to be too much and not satisfactory. We refine our model, getting the feedback from DM with a threshold value of 15 percent deviation from the targeted cost value. This time, the achieved levels for our designated goals are observed as follows: 8 for the total number of bases, 4937910 (km) for total flight distance; 835 for total unairworthy days, and $11501920 for total cost. The resulting

**Table 13. Refined solution results in the case of the earthquake scenario.**

Minimum Total Weighted Goal Deviation = 32.9%

| Number of Selected Bases | Selected Bases | Allocated Disaster Activity Zones | Flight Missions Planned |
|---|---|---|---|
| 8 | 4 | Zone 7 | 360 |
| | | Zone 14 | 360 |
| | 9 | Zone 3 | 112 |
| | | Zone 5 | 720 |
| | | Zone 15 | 76 |
| | 10 | Zone 2 | 720 |
| | | Zone 15 | 284 |
| | 11 | Zone 8 | 216 |
| | | Zone 10 | 360 |
| | | Zone 11 | 272 |
| | 18 | Zone 1 | 720 |
| | | Zone 3 | 16 |
| | | Zone 13 | 360 |
| | 19 | Zone 4 | 140 |
| | | Zone 6 | 720 |
| | 21 | Zone 3 | 592 |
| | | Zone 11 | 88 |
| | | Zone 12 | 360 |
| | 23 | Zone 4 | 580 |
| | | Zone 8 | 144 |
| | | Zone 9 | 360 |

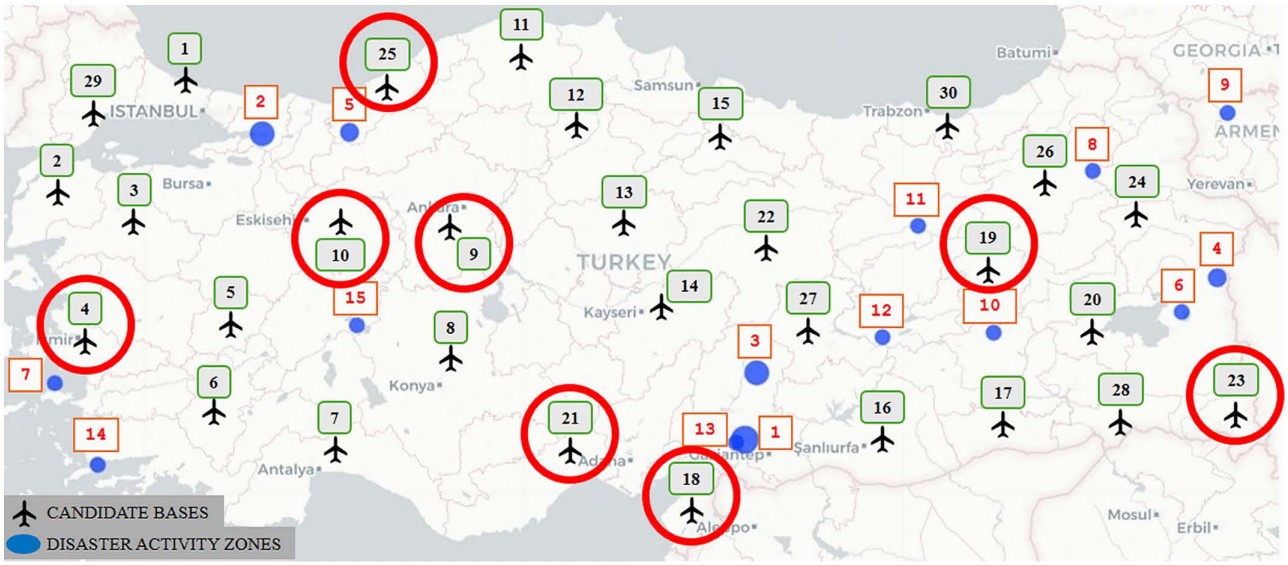

**Fig 11. Refined optimal base locations in the case of the earthquake scenario.**

**Table 14. Refined targeted and achieved levels of goals in the case of the earthquake scenario.**

| Objectives/Goals | Targeted Level | Achieved Level | Deviation in Targeted Level | Percentage Deviation in Targeted Level |
|---|---|---|---|---|
| Number of Bases Opened | 7 | 8 | 1 | 14.3% |
| Total Flight Distance (km) | 717020 | 914236 | 197216 | 27.5% |
| Total Unairworthy Days | 652 | 909 | 257 | 39.4% |
| Total Cost (dollars) | 7777181 | 9811979 | 2034798 | 26.2% |

percentage deviations are 0, 22, 2.2, and 8.7, respectively. A remarkable 8.57% improvement is observed in the cost value with a decrease of $1078970 from $12580890 to $11501920, even though there are some slight increases in other deviational values.

In the second scenario (Case 2), DM deems the deviation value of 18.4% in Goal 2, total flight distance, not satisfactory. Thus, DM sets a threshold value of 15% for the deviation in Goal 2. The achieved goal levels for the number of bases, total flight distance, total unairworthy days, and total cost turn into 9, 4937910, 835, and 11501920, respectively. The associated percentage deviations from targeted levels become 12.5, 9.5, 16.4, and 21.9. There is a noticeable 7.54% improvement in the distance value when compared to that of the original Model 1 results, a decrease of 361570 km from 4792590 to 4431020

In our last scenario (Case 3), DM is not satisfied with the deviation value of 18.4% in Goal 2, total flight distance, and with the deviation value of 18.8% in Goal 4, total cost, as well. Thus, DM sets a threshold value of 15% for both deviations in Goal 2 and Goal 4. The achieved goal levels for the number of bases, total flight distance, total unairworthy days, and total cost turn into 9, 4425060, 977, and 11752530, respectively. The associated percentage deviations from targeted levels become 12.5, 9.4, 19.6, and 11.1. The improvements we get for the distance and cost values are 7.66% and 6.58% respectively, when compared to the results of the original Model 1. We observe a decrease with a value of 367530 km from 4792590 to 4425060 in the flight distance and a decrease with a value of $ 828360 from 12580890 to 11752530 in the cost value.

Finally, to demonstrate the model's practical applicability, we present a real-world scenario regarding the case of an earthquake in Türkiye with an observed improvement of 5% in total flight distance value and 14.8% improvement in cost value.

As a result, the proposed multi-criteria decision-making model with DM in the loop provides solutions for UAV base locations and deployment plans with remarkable improvements when compared to the classical goal programming model. The results are potentially more satisfactory for DM and better reflect the real-life aspects of the problem since the interactive feedback from DM is integrated into the model, narrowing the gap between theory and practice.

Since this study focuses on the long-term planning of UAV operations, with particular emphasis on the selection and designation of appropriate bases, a study emphasizing short-term planning—accounting for daily weather conditions, topography, route patterns, and time scheduling aspects—would be valuable as future research.

## Author contributions

**Conceptualization:** Mustafa Erdem Bakir, Fatih Kasimoglu.

**Data curation:** Mustafa Erdem Bakir.

**Formal analysis:** Mustafa Erdem Bakir, Fatih Kasimoglu.

**Investigation:** Mustafa Erdem Bakir, Fatih Kasimoglu.

**Methodology:** Mustafa Erdem Bakir, Fatih Kasimoglu.

**Software:** Mustafa Erdem Bakir, Fatih Kasimoglu.

**Supervision:** Fatih Kasimoglu.

**Validation:** Mustafa Erdem Bakir, Fatih Kasimoglu.

**Visualization:** Mustafa Erdem Bakir.

**Writing – original draft:** Mustafa Erdem Bakir, Fatih Kasimoglu.

**Writing – review & editing:** Mustafa Erdem Bakir, Fatih Kasimoglu.

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
