## [Decision Letter · Decision Letter 0]

3 Nov 2025

Dear Dr. Kasimoglu,

Thank you for submitting your manuscript to PLOS ONE. After careful consideration, we feel that it has merit but does not fully meet PLOS ONE’s publication criteria as it currently stands. Therefore, we invite you to submit a revised version of the manuscript that addresses the points raised during the review process. Please see the comments from two reviewers at the bottom of this email.

We look forward to receiving your revised manuscript.

Kind regards,

Xiaoyong Sun

Academic Editor

PLOS ONE

Journal Requirements:

Reviewer's Responses to Questions

**Comments to the Author**

1. Is the manuscript technically sound, and do the data support the conclusions?

Reviewer #1: Partly

Reviewer #2: Partly

2. Has the statistical analysis been performed appropriately and rigorously?

Reviewer #1: Yes

Reviewer #2: N/A

3. Have the authors made all data underlying the findings in their manuscript fully available?

Reviewer #1: Yes

Reviewer #2: Yes

4. Is the manuscript presented in an intelligible fashion and written in standard English?

Reviewer #1: Yes

Reviewer #2: Yes

Reviewer #1: This manuscript presents an integrated Multi-Criteria Decision-Making (MCDM) model that combines the Analytic Hierarchy Process (AHP) with a Goal Programming (GP) optimization model to solve the problem of long-term UAV base location selection and mission planning for disaster management. The key novelty lies in the interactive feedback loop, where a Decision Maker (DM) can refine the model by setting threshold deviations for unsatisfactory goals, leading to more validated and satisfactory solutions.

Well-Structured Framework: The core contribution is the clear, step-by-step integration of a Multi-Attribute Decision Making (MADM) method (AHP) with a Multi-Objective Decision Making (MODM) method (Goal Programming) in an interactive loop. This is a non-trivial and valuable methodological proposal.

Addresses a Critical Gap: The paper correctly identifies a gap in the literature: the lack of integrating the Decision Maker's (DM) preferences during the solution process and not just at the initial weighting stage. The proposed "refinement" cycle (Algorithm 2) is a direct and elegant solution to this problem.

Generalizable Methodology: The proposed framework is not limited to UAV base planning. It can be adapted to any complex location-allocation problem with multiple, conflicting objectives and a stakeholder who needs to be actively involved (e.g., hospital location, logistics hub planning, emergency facility placement). This enhances its theoretical value.

Clarity and Reproducibility: The model formulation, algorithms, and process flow (Figure 2) are described with sufficient clarity for other researchers to understand, replicate, and build upon the method. The availability of code (GitHub) further strengthens this point.

Comments 1: While the integration is well-executed, the individual components (AHP, GP) are well-established. However, this is an incremental rather than a groundbreaking theoretical advances.

The choice of the four specific goals (number of bases, distance, unairworthy days, cost) is presented without a strong theoretical or literature-based justification for why this specific set is the most critical. A stronger theoretical paper would defend this choice more rigorously.

Comments 2: I have a big doubt on how the study’s model is practically applicable in the area where topography is very crucial. Please provide how the model’s specific assumptions are made with the practical consideration. Kindly address the following critical observations.

1. Line-of-Sight (LoS) and Signal Reliability: UAVs, especially for BVLOS (Beyond Visual Line of Sight) operations, often rely on radio or satellite communication. Mountains, hills, and urban canyons can block these signals, making a location that is geographically "close" in terms of distance completely unusable. The paper's model only considers a flat, 2D distance (R_ij), which is a massive oversimplification.

2. Take-off and Landing Requirements: A candidate base must have a sufficiently large, flat, and clear area for safe take-off and landing. A location that is optimal on a map might be on a steep slope, in a dense forest, or in an urban area with no clear space.

3. Wind Patterns and Microclimates: Topography dramatically influences local weather. A valley might be prone to fog, a mountain pass to high winds, and a coastal area to salt spray. The paper's single metric of "unairworthy days" (UAD_i) is too coarse; it doesn't capture how topography creates localized, persistent hazardous conditions that would make a specific site unsuitable, regardless of the regional weather data.

4. Flight Path Obstacles: The optimal path from a base to a Disaster Activity Zone (DAZ) is not a straight line. It must navigate around terrain features. The paper's "flight distance" is a simple straight-line radius, ignoring that the actual flight path might be much longer and more energy-consuming to go around a mountain range.

5. Is it around 5 km radius of coverage in your model???

Comments 3: In several key places, the paper uses a comma (,) as a decimal separator for the objective function value, which is a percentage.

Critical Examples:

Optimal Objective Function Value ... = 7,2% (Page 21, Table 3)

… = 7,8% (Page 24, Table 5)

… = 14,4% and ... = 15,6% (Page 26 & 28, Tables 7 & 9)

Number formats for km and dollars are confusing.

Page 16…..total unairworthy days, and total cost turn out to be 8, 4.792.590, 833, and 12.580.890

Page 16 …… Total Flight Distance (km) in table 4….4.046.475 ……….. 4.792.590

Page 16…….. Total Cost (dollars) in table 4…… 10.582.690…… 12.580.890….. 1.998.197

Check the entire manuscript or it is a coding?

Reviewer #2: 1. Empirical Data Validation: The study relies on simulated (“generic”) data. While this is acceptable for methodological development, the absence of real-world or semi-real data limits external validity.

Suggestion: Incorporate or reference a case study using actual disaster-region data (e.g., UAV deployment during forest fires, floods, or earthquakes in Turkey) to demonstrate the model’s practical applicability.

2. Figures 3–6 are informative but could be enhanced with color or clearer legends to help distinguish bases and disaster activity zones visually.

3. Consider including a comparative chart showing percentage improvements across all cases (1–3) to make the improvements easier to interpret.

**Do you want your identity to be public for this peer review?** For information about this choice, including consent withdrawal, please see our Privacy Policy

Reviewer #1: No

Reviewer #2: No

---

## [Author Response · Author response to Decision Letter 1]

1 Dec 2025

We have carefully incorporated the reviewers’ suggestions, as detailed in the response-to-reviewers file, and as a result, a substantially improved manuscript has been produced thanks to the constructive and insightful feedback provided by the reviewers.

---

## [Decision Letter · Decision Letter 1]

18 Dec 2025

An Integrated Multi-Criteria Decision-Making Model for Long-Term Planning of UAVs in Disaster Management

PONE-D-25-47729R1

Dear Dr. Kasimoglu,

We’re pleased to inform you that your manuscript has been judged scientifically suitable for publication and will be formally accepted for publication once it meets all outstanding technical requirements.

Kind regards,

Xiaoyong Sun

Academic Editor

PLOS One

sunx1@sdau.edu.cn

Additional Editor Comments (optional):

Reviewers' comments:

Reviewer's Responses to Questions

**Comments to the Author**

Reviewer #1: All comments have been addressed

2. Is the manuscript technically sound, and do the data support the conclusions?

Reviewer #1: Yes

3. Has the statistical analysis been performed appropriately and rigorously?

Reviewer #1: Yes

4. Have the authors made all data underlying the findings in their manuscript fully available?

Reviewer #1: Yes

5. Is the manuscript presented in an intelligible fashion and written in standard English?

Reviewer #1: Yes

Reviewer #1: Comments:

Thank you for submitting the revised version of your manuscript, "An Integrated Multi-Criteria Decision-Making Model for Long-Term Planning of UAVs in Disaster Management." I have carefully reviewed the changes made in response to the previous round of comments.

The authors have successfully addressed the key concerns raised in the first review. In particular, section 3.1 p.8. has been revised broadly to capture key comments raised by incorporating operational assumptions that can benefit the readers. Accordingly, conclusion of the research show clear future scope based on whatever the theoretical contribution that the paper demonstrates. In addition, comment 3 has been taken care throughout the manuscript.

Thank you and best wishes!

**Do you want your identity to be public for this peer review?** For information about this choice, including consent withdrawal, please see our Privacy Policy

Reviewer #1: No

---

## [Editor Report · Acceptance letter]

PONE-D-25-47729R1

PLOS One

Dear Dr. Kasimoglu,

I'm pleased to inform you that your manuscript has been deemed suitable for publication in PLOS One. Congratulations! Your manuscript is now being handed over to our production team.

Kind regards,

on behalf of

Dr. Xiaoyong Sun

Academic Editor

PLOS One